# Antimicrobial Growth Promoters Altered the Function but Not the Structure of Enteric Bacterial Communities in Broiler Chicks ± Microbiota Transplantation

**DOI:** 10.3390/ani13060997

**Published:** 2023-03-09

**Authors:** Colten R. Hodak, Danisa M. Bescucci, Karen Shamash, Laisa C. Kelly, Tony Montina, Paul B. Savage, G. Douglas Inglis

**Affiliations:** 1Lethbridge Research and Development Centre, Agriculture and Agri-Food Canada, Lethbridge, AB T1J 4B1, Canada; 2Department of Chemistry and Biochemistry, University of Lethbridge, Lethbridge, AB T1K 3M4, Canada; 3Southern Alberta Genome Sciences Centre, University of Lethbridge, Lethbridge, AB T1K 3M4, Canada; 4Department of Chemistry and Biochemistry, Brigham Young University, Provo, UT 84602, USA

**Keywords:** virginiamycin, ceragenin, CSA-44, metabolomics, metabolome

## Abstract

**Simple Summary:**

The use of antimicrobial growth promoters (AGPs) is no longer allowed in livestock production in many jurisdictions globally due to the potential emergence of antimicrobial resistance in zoonotic bacteria. Understanding the mode of action of AGPs may aid in the development of effective alternatives, and we examined the impact of the conventional antibiotic AGP, virginiamycin, and an AGP alternative, ceragenin (CSA-44), on the structure and function of the intestinal microbiota in young broiler chickens. Additionally, the temporal establishment of intestinal bacterial communities ± administration of virginiamycin and CSA-44 was evaluated by transplanting microbiota from healthy adult donor chickens into 1-day-old chicks. Microbiota transplantation showed an early establishment of a stable and diverse bacterial community. Neither virginiamycin nor CSA-44 impacted bird growth or the structure of bacterial communities regardless of microbiota transplantation. However, the complexity of the intestinal bacterial community in birds administered virginiamycin and CSA-44 affected the quantity and type of metabolites produced. Study findings demonstrate that the evaluation of effective AGP alternatives must comprehensively address their effects on the host–microbiota interaction, including impacts on the function of the intestinal microbiota.

**Abstract:**

Non-antibiotic alternatives to antimicrobial growth promoters (AGPs) are required, and understanding the mode of action of AGPs may facilitate the development of effective alternatives. The temporal impact of the conventional antibiotic AGP, virginiamycin, and an AGP alternative, ceragenin (CSA-44), on the structure and function of the broiler chicken cecal microbiota was determined using next-generation sequencing and ^1^H-nuclear magnetic resonance spectroscopy (NMR)-based metabolomics. To elucidate the impact of enteric bacterial diversity, oral transplantation (±) of cecal digesta into 1-day-old chicks was conducted. Microbiota transplantation resulted in the establishment of a highly diverse cecal microbiota in recipient chicks that did not change between day 10 and day 15 post-hatch. Neither virginiamycin nor CSA-44 influenced feed consumption, weight gain, or feed conversion ratio, and did not affect the structure of the cecal microbiota in chicks possessing a low or high diversity enteric microbiota. However, metabolomic analysis of the cecal contents showed that the metabolome of cecal digesta was affected in birds administered virginiamycin and CSA-44 as a function of bacterial community diversity. As revealed by metabolomics, glycolysis-related metabolites and amino acid synthesis pathways were impacted by virginiamycin and CSA-44. Thus, the administration of AGPs did not influence bacterial community structure but did alter the function of enteric bacterial communities. Hence, alterations to the functioning of the enteric microbiota in chickens may be the mechanism by which AGPs impart beneficial health benefits, and this possibility should be examined in future research.

## 1. Introduction

The enteric microbiota plays an essential role in host health by participating in nutrient uptake, contributing to immune development, and directly or indirectly competing with pathogens [1]. Chickens in commercial production differ from other animals as neonatal chicks possess a very low diversity enteric bacterial community due to the strict hygiene practices in hatcheries and limited exposure to maternal microbiota. A low-diversity microbiota increases the susceptibility of chicks to pathogenic diseases [2]. The establishment of an autochthonous microbiota starts in the first days of life, with many different factors contributing to the shaping of the bacterial community in the gastrointestinal tract, including exposure to extrinsic and intrinsic factors (i.e., food, water, bedding, gender, age, and breed, among others) [3,4]. It has been suggested that the composition of the microbiota, including instances where the community structure varies amongst birds, may influence the metabolome, affecting bird nutrition and response to various treatments. Cecal microbiota transplants applied to egg surfaces have been shown to reduce variation of bacterial composition among birds, although they did not reduce the feed conversion ratio [5]. Additionally, the administration of cecal microbiota from healthy adult donors to day-old broiler chicks prevented disease by increasing bacterial diversity and promoting positive immune responses [2]. The effect of cecal microbiota transplantation on growth promotion has yet to be comprehensively examined.

Historically, the administration of antibiotics at non-therapeutic concentrations has been used to optimize livestock production (i.e., as antimicrobial growth promoters) [6]. In this regard, antimicrobial growth promoters (AGPs) have been used in chickens to prevent diseases such as necrotic enteritis incited by *Clostridium perfringens* [7]. Moreover, AGPs may enhance animal growth and performance [8], although the benefits achieved are variable. Currently, the mechanisms by which AGPs function are not fully understood and elucidating the mode of action of AGPs may facilitate the development of effective non-antibiotic alternatives [9]. One of the proposed mechanisms by which AGPs function is via the modulation of enteric bacterial communities resulting in the competitive exclusion of pathogens [9]. However, the dynamic environment of the intestine and the high variability in the structure of the enteric microbiota among individual animals represent significant challenges in elucidating mechanisms by which AGPs function. A salient issue facing AGP use in livestock is the potential for the selection of enteric bacteria resistant to antibiotics and subsequent transmission of zoonotic bacteria resistant to medically important antibiotics to people. This has resulted in restrictions on AGP use in many countries, including Canada [10]. Restrictions on the use of AGPs has had economic consequences for chicken production, including decreased growth rates and feed efficiency, higher morbidity and mortality rates, and increased veterinary costs resulting from increased therapeutic treatment of diseases, which have resulted in an increase in the price of meat [11]. Thus, there is a pressing need to identify and develop effective alternatives to antibiotic AGPs [12].

Considerable effort is currently being expended to identify non-antibiotic alternatives to antibiotic AGPs, including the identification of natural products of the immune system that can be used to enhance intestinal defenses and overall animal production health. Antimicrobial peptides (AMPs) are present in birds from embryonic stages and develop continuously during their lifetime. AMPs include β-defensins, cathelicidins, lysozymes, cationic proteins, and polypeptides that are naturally produced throughout the gastrointestinal tract of chickens [13]. The supplementation of feed with AMPs has been shown to improve growth performance by activating immune cells, enhancing intestinal morphology, and establishing healthy commensal bacteria by reducing intestinal pathogens [13,14]. Within the AMPs, cathelicidins have been of particular interest due to their highly cationic characteristic [13], which is directly related to their effectiveness against microorganisms, suggesting that membrane disruption is the primary mechanism attributed to their antimicrobial properties [15]. The use of AMPs as an alternative to antibiotics has been previously proposed [13]. However, synthesizing AMPs for commercial-scale use presents several problems, including the complexity and expense of peptide production, the salt sensitivity of AMP activity, and their susceptibility to proteolytic degradation [16]. Ceragenins, as non-peptide mimics of cathelicidins, are not sensitive to enzymatic degradation and can be effectively delivered orally to the gastrointestinal tract without loss of activity [17]. Ceragenins have been shown to possess antimicrobial activities [18,19,20]; however, their impacts on growth promotion in chickens are currently unknown.

A primary purpose of the study was to obtain information on the potential mechanisms by which AGPs may provide a health benefit to broilers. In this regard, we investigated the effects of the conventional AGP, virginiamycin, and an alternative AGP candidate (i.e., the ceragenin, CSA-44) on the structure and function of the cecal microbiota. As the use of AMPs, including AMP mimics, is an active area of research in human medicine, we chose to examine the impact of the cathelicidin mimic, CSA-44 [21], as a potential growth promoter in chickens. We have previously shown that the early establishment of a diverse microbiota in chicks has profound impacts on the health of the birds [2], and microbiota transplantation was included as an experimental variable to address the impact of the early establishment of a diverse and homogeneous microbiota on the AGP-host-microbiota interaction. We hypothesized that the administration of virginiamycin and CSA-44 will modify the structure and function of the intestinal microbiota to enhance broiler chick growth. Additionally, the administration of complex microbiota to neonatal chicks in a microbiologically controlled environment will result in a homogeneous, diverse, and robust enteric microbiota, thereby promoting growth. We chose to maintain broiler chicks in a microbiologically controlled environment to facilitate the interpretation of AGP effects on the enteric microbiota. To test the above hypotheses, newly hatched chicks were orally transplanted with a cecal microbiota obtained from healthy adult donor birds and administered virginiamycin or CSA-44 in feed. Characterization of the microbiota structure and function were conducted using next-generation sequencing and ^1^H-nuclear magnetic resonance spectroscopy (NMR)-based metabolomic analyses, respectively.

## 2. Materials and Methods

### 2.1. Experimental Design

The experiment was designed with two levels of microbiota transplant (±MT), three levels of diet treatment (Control, CSA-44, and virginiamycin), and two levels of sampling time (day 10 and day 15 post-hatch endpoints) arranged as a completely randomized design, with four birds (replicates) per treatment (Appendix A). The experiment was repeated on two occasions (i.e., “runs”) with two–four replicates per treatment included in each run (48 animals total).

### 2.2. Animals and Husbandry

On the morning of the hatch, Ross 308FF broiler chicks were obtained from a local hatchery (Lethbridge, AB, Canada). Chicks were transported to the Agriculture and Agri-Food Canada (AAFC) Lethbridge Research and Development Centre (LeRDC). To prevent the introduction of microorganisms during transport, animals were placed in a sterile container fully wrapped with autoclaved surgical drapes. Transport from the hatchery to LeRDC took ≈5 min. Immediately upon arrival at LeRDC, the sterile transport container was placed in an operating biosafety cabinet (BSC), the transport drapes were removed, and the chicks were individually weighed taking care not to introduce microorganisms from the local environment. Chicks were then randomly transferred into a sterile Techniplast (Techniplast, Montreal, QC, Canada) individually ventilated cage (IVC) within the BSC. Immediately following the placement of the chicks in an IVC, they were placed in a cage rack operated in containment mode (Techniplast). Sterile wood bedding was provided, and cage changes were conducted every other day. A custom starter diet (Appendix A) and water were provided *ad libitum*. Ambient temperature, lighting, and humidity were maintained following the Canadian Council on Animal Care Guidelines [22]. Birds were maintained at 30 °C for the first 2 days post-hatch, 28 °C for the next 2 days, and then maintained at 26 °C for the remainder of the experiment. Ambient humidity ranged from ≈40–50%. Birds were maintained on an 18 h light: 6 h dark cycle throughout the experimental period. To minimize stress, two replicate chicks were housed together per cage. Feed consumption and bird weight were recorded daily.

### 2.3. Microbiota Transplantation

Cecal digesta was obtained from six healthy adult broilers of known health status. Birds were humanely sacrificed at the barn and immediately transported to AAFC LeRDC for processing. Feathers were removed, and the skin was sanitized using 70% ethanol. A laparotomy was performed, and ceca were exposed and double-ligated (proximal ends) using sterile zip ties and then harvested by making an incision between the two ligatures (situated ≈10 mm from each other). Ceca were placed in a sterile plastic tray and transferred into a Thermo Forma 1025 anaerobic chamber (Thermo Fisher Scientific Inc., Waltham, MA, USA) containing a nitrogen-predominant atmosphere (85:5:10% N_2_:H_2_:CO_2_); care was taken to ensure that the surface of the ceca were not contaminated with digesta. Within the anaerobic chamber, individual ceca were opened longitudinally with sterile scissors, and digesta was removed and placed into a sterile urine sample container, weighed, and suspended in a degassed solution of Columbia Broth (CB; VWR, Mississauga, ON, Canada) with 40% (*v*/*v*) glycerol (Fisher Scientific, Ottawa, ON, Canada) and L-cysteine (1.0 g/L; Fisher Scientific) at a ratio of one (digesta) to three (CB-glycerol) (*v*/*v*). This process was repeated for each of the six donor birds, and slurries from all the birds were combined. The combined slurry was then dispensed into 2 mL screw cap tubes, and the sealed tubes were removed from the anaerobic chamber and stored at −80 °C until used. On the day of the transplantation (1-day-old chicks), the frozen digesta slurry in tubes was thawed in the anaerobic chamber at 37 °C, and 200 µL aliquots were transferred into 1 mL sterile syringes (Soft Pack; Western Drug Distribution Center, Edmonton, AB, Canada). MT+ treatment chicks were gavaged 1 day after arrival at LeRDC (Appendix A). Briefly, a 17-gauge, 5-cm-long gavage needle (Cadence Science. Inc., Cranston, RI, USA) fitted onto a syringe was inserted into the esophagus of a restrained chick, and the digesta slurry (200 µL) was slowly discharged. Control treatment birds (i.e., MT−) were gavaged with 200 µL of sterile CB-glycerol.

### 2.4. Antimicrobial Growth Promoter Treatments

On day 3, randomly selected chicks were assigned to one of the three following diet treatments: (1) unamended starter ration (control treatment) [2]; (2) starter ration containing 20 mg/kg virginiamycin (Phibro, Teaneck, NJ, USA); and (3) starter ration containing 620 mg/kg of synthesized CSA-44 (CSA treatment) [21] (Appendix A). Diet treatments were custom formulated at the AAFC LeRDC Feed Mill.

### 2.5. Sample Collection

Recently excreted feces were collected on days 5, 10, and 15 for short-chain fatty acid (SCFA) analysis. On days 10 and 15, the animals were anesthetized with isoflurane and humanely euthanized via cervical dislocation. A laparotomy was performed, the entire intestine was removed, ceca were longitudinally opened, and cecal digesta was immediately collected and stored at −80 °C until processed (i.e., for characterization of the bacterial communities and metabolomics).

### 2.6. Short Chain Fatty Acid Quantification

Concentrations of SCFAs in feces were determined as previously described [23]. Briefly, recently excreted feces (i.e., 600–800 mg) were collected at 5, 10, and 15 days post-hatch and kept at room temperature for a maximum of 30 min. Feces homogenized in phosphate-buffered saline (pH 7.2) at 1:1 (*w*/*v*) and 25% meta-phosphoric acid (Sigma Aldrich, Oakville, ON, Canada) was added to the homogenate at a 1:4 (*v*/*v*) ratio. Samples were vortexed for 30 s and then centrifuged for 75 min at 16,000× *g*. Supernatants were collected and stored at −20 °C until analyzed. A gas chromatograph (Model 6890N with 7683 Series Injector; Agilent Technologies Canada Inc., Mississauga, ON, Canada) was used to determine concentrations of acetic, propionic, isobutyric, butyric, isovaleric, valeric, and caproic acid, as previously described [24].

### 2.7. Bacterial Community Characterization

Bacterial DNA was extracted from cecal digesta using the QIAamp Power Fecal Pro kit (Qiagen Inc., Toronto, ON, Canada) following the manufacturer’s protocol. Extracted DNA was processed following a dual-indexed sequencing strategy for creating 16S rRNA libraries in the V4 region. Extracted genomic DNA was first diluted (≈20–50 ng) and then amplified with Illumina-indexed adaptor primers (V4 Schloss primers) [25]. Each PCR reaction contained 12.5 µL Paq5000 Master Mix (Agilent Technologies, Mississauga, ON, Canada), 1 µL of each 10 µM primer (IDT, Coralville, IA, USA), 8.5 µL of nuclease-free water (Qiagen Inc.), and 2 µL of the diluted template DNA. The PCR reaction was run on an Eppendorf Mastercycler Pro S thermocycler (Eppendorf Canada Ltd., Mississauga, ON, Canada), with cycle conditions of 95 °C for 2 min, followed by 25 cycles of 95 °C for 20 s, 55 °C for 15 s, and 72 °C for 5 min, and one cycle at 72 °C for 10 min. The final amplicon length was ≈385 bp. Amplicons were purified with 24 µL of AMPure XP beads (Beckman Coulter Canada Inc., Mississauga, ON, Canada) following the manufacturer’s protocol. The efficacy of the purification step was confirmed by agarose gel electrophoresis, followed by quantification with a Qubit 4 fluorometer (Life Technologies Inc., Burlington, ON, Canada). The concentration of DNA was calculated in nM based on the size of the amplicons. Indexed DNA libraries were normalized to 1.5 ng/µL and were pooled. PhiX control DNA (25% *v*/*v*; Illumina, San Diego, CA, USA) was run with the normalized DNA library, and both were denatured and diluted to 3 pM prior to loading into the MiSeq v2 (500 cycle) reagent cartridge (Illumina). Sequences were obtained using a MiSeq benchtop sequencer (Illumina) located at the AAFC LeRDC.

### 2.8. Characterization of the Metabolome

Cecal digesta was weighted to ≈150 mg. Metabolomics buffer (0.125 M KH_2_PO_4_, 0.00375 M NaN_3_, and 0.375 M KF; pH 7.4) was added to the cecal digesta samples at a ratio of 2:1 (*v*/*w*) and homogenized using a Bullet Blender tissue homogenizer (Next Advance; Troy, NY, USA) with 150 mg of 2-mm-diameter zirconium oxide beads (Next Advance) at setting eight for 5 min. Samples were centrifuged for 5 min at 14,000× *g*, and supernatants were filtered through a 3000 MWCO Amicon Ultra-0.5 filter (Millipore Sigma, Oakville, ON, Canada) by centrifugation at 14,000× *g* for 30 min at 4 °C. An aliquot of 360 µL was mixed with 200 µL of metabolomics buffer and 140 µL of deuterium oxide containing 0.05% (*v*/*v*) trimethylsilylpropanoic acid (TSP) for a final volume of 700 µL; TSP was used as a chemical shift reference for ^1^H-NMR spectroscopy. The solutions were vortexed and centrifuged at 12,000× *g* for 5 min at 4 °C, and an aliquot of 550 µL of the supernatant of each sample was loaded in a 5 mm NMR tube. Spectra were obtained as previously described [26].

### 2.9. Statistical Analyses

Analyses of feed consumption, bird growth, and feed conversion ratio averaged over time were conducted using the mixed procedure of SAS (SAS Institute Inc., Cary, NC, USA). The two endpoint variables were combined (*n* = eight replicate birds), and ‘run’ was included as a random effect. In the event of a significant main or interaction effect, the least squares means (lsmeans) test was used to compare treatments.

Analysis of 16S rRNA sequence data was conducted in QIIME2 (version 2022.8) [27]. Raw reads were denoised with DADA2 [28]. A phylogenetic tree of amplicon sequence variants (ASVs) was generated with taxonomy classification conducted with the reference SILVA 138 database (silva-138-99-nb-classifier.qza). Alpha-diversity (i.e., Shannon’s index) and β-diversity (i.e., Bray–Curtis) were evaluated with QIIME2. Group differences in α-diversity and β-diversity were evaluated by Kruskal–Wallis and permutation permanova tests, respectively. Differential abundance of specific families was conducted with ANCOM [29].

Metabolomics data were analyzed as described previously [30]. Briefly, metabolite spectral bins were subjected to both univariate and multivariate analyses to determine which metabolites were significantly altered between treatments using MATLAB (MathWorks, MA, USA). The univariate measures were calculated using a decision tree algorithm as described by Goodpaster et al. [31]. MATLAB was used to calculate the percent difference of the bins between treatments, and the R package, MetaboanalystR [30], was used to carry out unsupervised principal component analysis (PCA). Metabolites were identified using Chenomx 8.2 NMR Suite (Chenomx Inc., Edmonton, AB, Canada).

## 3. Results

### 3.1. Administration of Virginiamycin or Ceragenin CSA-44 Did Not Appreciably Affect Feed Consumption, Body Weight Gain, or Feed Conversion Ratio

Averaged over the 15-day experimental period, the administration of virginiamycin at a non-therapeutic dose of 20 mg/kg had no effect (*p* ≥ 0.067) on the feed consumption, body weight gain, or feed conversion ratio relative to control treatment birds regardless of the microbiota transplantation (Appendix A). Relative to the control treatment, birds administered CSA-44 consumed more diet (*p* < 0.001) (Appendix A). However, average daily weight gain was reduced (*p* ≤ 0.051) in CSA treatment birds that received the microbiota transplant (Appendix A). The administration of CSA-44 did not affect (*p* = 0.320) the feed conversion ratio relative to the control treatment (Appendix A).

### 3.2. Cecal Microbiota Transplantation Altered the Structure of the Cecal Bacterial Community

The composition and diversity of cecal bacteria differed (*p* ≤ 0.002) between birds that were administered the microbiota transplant (MT+) relative to chicks administered CB-glycerol alone (MT−) (Figure 1A,B and Figure 2A,B). At both the day 10 and 15 endpoints, bacterial communities in ceca of birds not administered the microbiota transplant were mainly comprised of bacteria within the families *Lachnospiraceae* (60%), *Ruminococcaceae* (12%), and *Lactobacillaceae* (10%). Although birds administered the microbiota transplant also contained a high relative abundance of bacteria in these three families (i.e., 33%, 11%, and 3%, respectively), bacteria in other families were also abundant. In this regard, bacteria in the families *Bacteroidaceae*, *Rikenellaceae*, *Sutterellaceae*, and *Atopobiaceae* were only observed in the cecal digesta of chicks administered the microbiota transplant. Additionally, temporal differences in the relative abundance of bacterial taxa between birds administered the microbiota transplant were observed. In this regard, a higher abundance of bacteria in the families *Butyricicoccaceae*, *Acidaminococcaceae*, *Sutterellaceae*, *Peptococcaceae*, and *Atopobiaceae* were observed in birds administered the microbiota transplant at the day 10 endpoint, whereas at the day 15 endpoint, only bacteria in the family *Enteroccocaceae* were more abundant (Figure 1A,B).

### 3.3. The Structure of the Cecal Bacterial Community Varied over Time but Only in Birds Not Administered the Microbiota Transplant

Both the α-diversity (*p* ≤ 0.033) and β-diversity (*p* ≤ 0.035) of cecal bacterial communities differed between the day 10 and day 15 endpoints in chicks that were not administered the microbiota transplant (Figure 2A,B). In contrast, there was no difference (*p* ≥ 0.092) in either α- or β-diversity between the two endpoints in chicks that were administered the microbiota transplant.

**Figure 1 animals-13-00997-f001:**
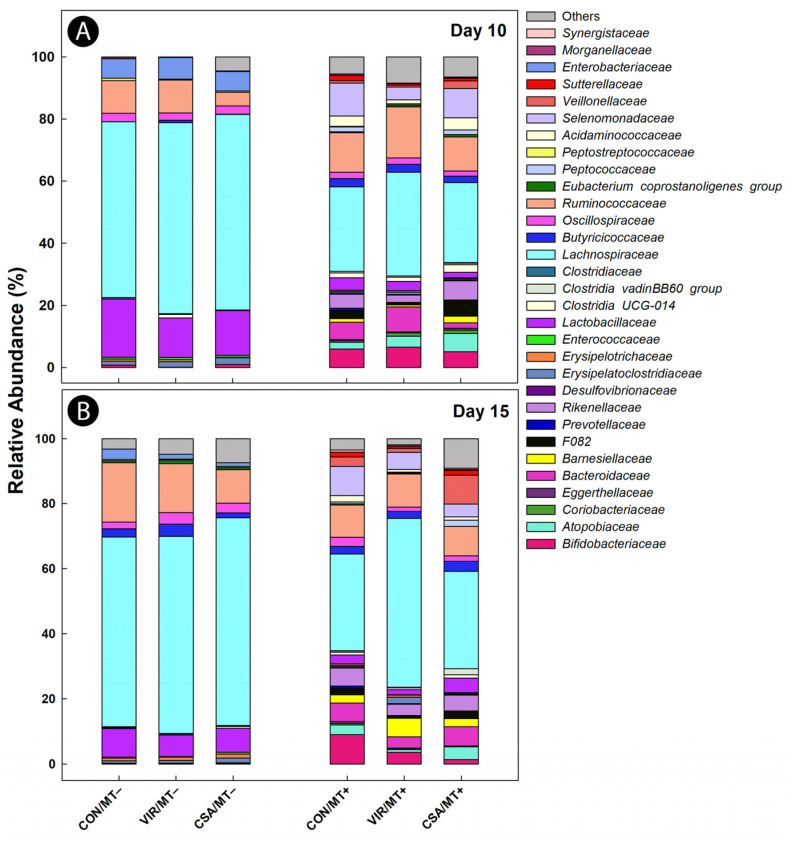
Relative abundance of bacterial communities in the cecal digesta of broiler chicks administered an unamended starter diet (CON), a diet supplemented with virginiamycin (VIR), or a diet supplemented with the ceragenin, CSA-44 (CSA). Birds received a microbiota transplant (MT+) or buffer alone (MT−) on day 1 post-hatch. (**A**) Day 10 post-hatch; (**B**) day 15 post-hatch.

### 3.4. Neither Microbiota Transplantation nor the Administration of Virginiamycin and Ceragenin CSA-44 Affected Intestinal Fermentation

Concentrations of acetic acid, propionic acid, butyric acid, isovaleric acid, valeric acid, and caproic acid were examined in chicken feces. Acetic acid was the only SCFA detected above the limit of detection in feces, and no differences (*p* ≥ 0.097) were found among treatments (data not shown).

### 3.5. The Structure of the Cecal Bacterial Community Was Minimally Impacted by Virginiamycin and Ceragenin CSA-44

The administration of virginiamycin or CSA-44 had no impact (*p* ≥ 0.084) on the temporal diversity of bacterial communities in chicks that were administered the microbiota transplant (Figure 2A,B). In contrast, α-diversity (*p* = 0.033) but not β-diversity (*p* = 0.068) was reduced in birds administered CSA-44 that were not administered the microbiota transplant at the day 15 endpoint (Figure 2A); this was attributed to the lower abundance of bacteria in the family *Butyricicoccaceae*.

**Figure 2 animals-13-00997-f002:**
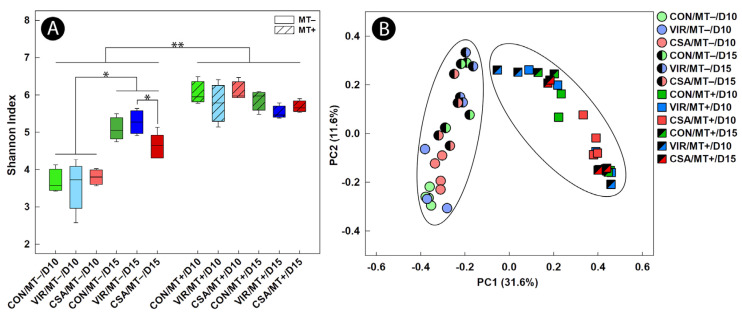
Diversity of bacteria in cecal digesta of broiler chicks administered an unamended starter diet (CON), a diet supplemented with virginiamycin (VIR), or a diet supplemented with the ceragenin, CSA-44 (CSA). Birds received a microbiota transplant (MT+) or buffer alone (MT−) on day 1 post-hatch. Samples were obtained 10 days post-hatch (D10) and 15 days post-hatch (D15). (**A**) Shannon index of α-diversity; (**B**) Bray–Curtis β-diversity. Ellipses in figure (**B**) distinguish MT– birds on the left and MT+ birds on the right. * *p* < 0.050. ** *p* < 0.010.

### 3.6. Microbiota Transplantation Altered the Function of the Cecal Digesta Microbiota

Metabolites were extracted from cecal digesta and analyzed by ^1^H-NMR to ascertain the impact of microbiota transplantation and the administration of virginiamycin or CSA-44 on the function of the cecal microbiota (i.e., at individual endpoints). Unsupervised PCA showed group separation of metabolite bins between birds administered the microbiota transplant at the day 10 endpoint for all three AGP treatments (Figure 3A–C). At the day 15 endpoint, unsupervised group separation was also observed in chicks administered the microbiota transplant in virginiamycin and control treatment birds but not in CSA treatment birds (Figure 3D–F). For control treatment birds at the day 10 endpoint, higher (*p* ≤ 0.045) concentrations of 2-isopropylmalic acid, alanine, and isoleucine were observed in chicks receiving the microbiota transplant (Figure 4). At the day 15 endpoint, control treatment birds administered the microbiota transplant showed an increase (*p* = 0.013) in the concentration of glucose-6-phosphate, and decreases (*p* ≤ 0.050) in the concentrations of glucose and glucose-1-phosphate.

### 3.7. Microbiota Transplantation Affected the Cecal Digesta Metabolome over Time

To determine the impacts of microbiota transplantation on the temporal function of the cecal microbiota, the metabolome of control treatment birds at the day 10 versus the day 15 endpoint was compared. Unsupervised group separation of metabolite bins was observed between the two endpoints in the control treatment regardless of the microbiota transplant treatment (Figure 5A,B). For control treatment birds not administered the microbiota transplant, a higher (*p* = 0.024) concentration of isoleucine, and lower (*p* ≤ 0.040) concentrations of 2-isopropylmalic acid, taurine, and tyrosine were observed at the day 15 relative to the day 10 endpoint (Figure 6). In control treatment birds administered the microbiota transplant, higher (*p* ≤ 0.043) concentrations of glutamate, isoleucine, and taurine, and lower (*p* ≤ 0.038) concentrations of glucose, glucose-1-phosphate, and leucine were observed at the day 15 relative to the day 10 endpoint.

**Figure 3 animals-13-00997-f003:**
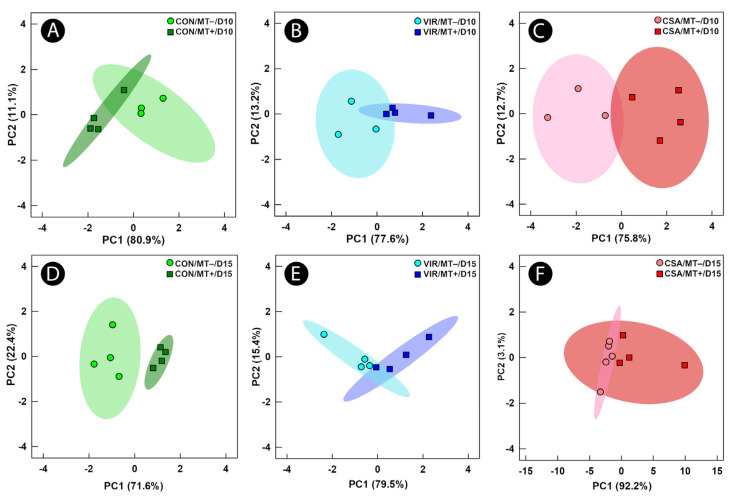
Principal component analysis score plots of metabolites bins in cecal digesta of broiler chicks administered an unamended starter diet (CON), a diet supplemented with virginiamycin (VIR), or a diet supplemented with the ceragenin, CSA-44 (CSA). Birds received a microbiota transplant (MT+) or buffer alone (MT−), and samples were obtained 10 days post-hatch (D10) and 15 days post-hatch (D15). Each marker represents one bird, and the shaded ellipses represent 95% confidence intervals. The x- and y-axes show principal components one and two, respectively, with the number in brackets indicating the percent variance of each component. Metabolite bins were determined to differ (*p* ≤ 0.050) according to a univariate Mann–Whitney U-test. (**A**) CON/MT– vs. CON/MT+ at day 10; (**B**) VIR/MT– vs. VIR/MT+ at day 10; (**C**) CSA/MT− vs. CSA/MT+ at day 10; (**D**) CON/MT– vs. CON/MT+ at day 15; (**E**) VIR/MT– vs. VIR/MT+ at day 15; (**F**) CSA/MT− vs. CSA/MT+ at day 15.

**Figure 4 animals-13-00997-f004:**
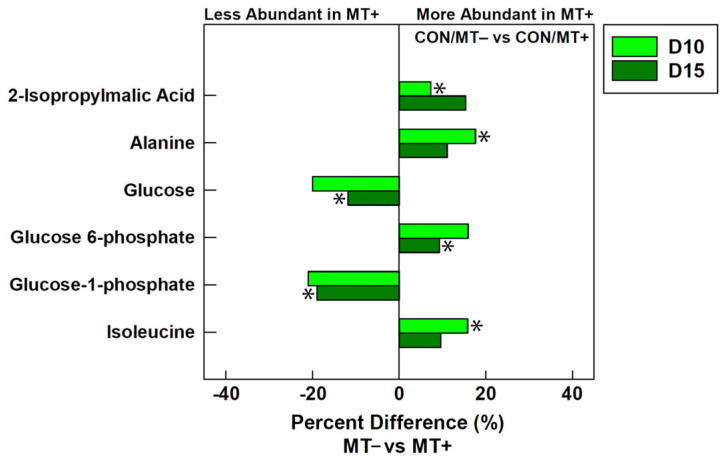
Average percent difference in discriminated metabolites between in cecal digesta of broiler chicks that were administered an unamended starter diet (CON) and received a microbiota transplant (MT+) or buffer alone (MT−). Samples were analyzed 10 days post-hatch (D10) and 15 days post-hatch (D15). Metabolites that were altered (*p* ≤ 0.050) according to univariate Mann–Whitney analysis are denoted with an asterisk.

**Figure 5 animals-13-00997-f005:**
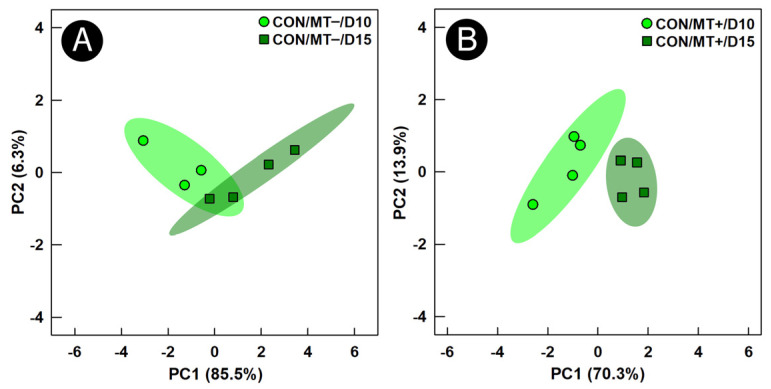
Principal component analysis score plots of metabolite bins in cecal digesta of broiler chicks administered an unamended starter diet (CON) and received a microbiota transplant (MT+) or buffer alone (MT−). Samples were analyzed 10 days post-hatch (D10) and 15 days post-hatch (D15). Each marker represents one bird, and the shaded ellipses represent 95% confidence intervals. The x- and y-axes show principal components one and two, respectively, with the number in brackets indicating the percent variance of each component. Metabolite bins were determined to differ (*p* ≤ 0.050) according to a Univariate Mann–Whitney U-test. (**A**) CON/MT– Day 10 vs. Day 15; (**B**) CON/MT+ Day 10 vs. Day 15.

**Figure 6 animals-13-00997-f006:**
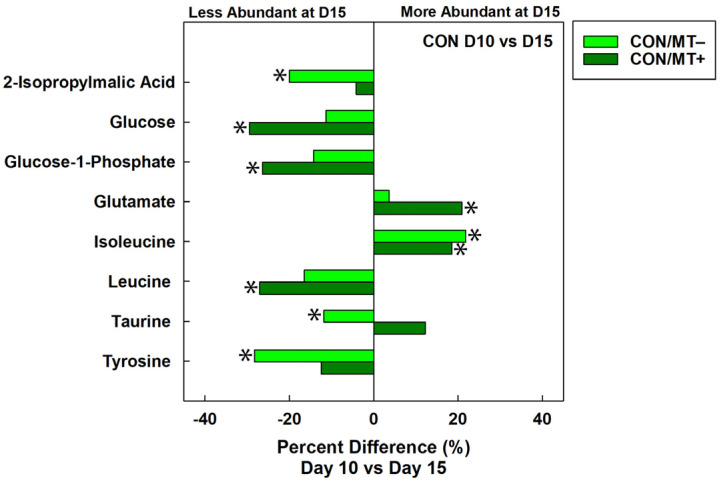
Average percent difference in discriminated metabolites in cecal digesta between broiler chicks that were administered an unamended starter diet (CON) and received a microbiota transplant (MT+) or buffer alone (MT−). Samples were analyzed 10 days post-hatch (D10) and 15 days post-hatch (D15). Metabolites that were altered (*p* ≤ 0.050) according to Univariate Mann–Whitney analysis are denoted with an asterisk.

### 3.8. Administration of Virginiamycin and Ceragenin CSA-44 Affected the Cecal Digesta Metabolome 

To determine the impacts of virginiamycin or CSA-44 administration on the temporal function of the cecal microbiota, cecal digesta metabolomes were contrasted with control treatment birds at the day 10 and day 15 endpoints. Unsupervised group separation of metabolite bins was observed between control and virginiamycin treatment birds, as well as between control and CSA treatment birds (Figure 7A–H).

For virginiamycin treatment birds not administered the microbiota transplant, higher (*p* ≤ 0.042) concentrations of phenylalanine and tyrosine, and lower (*p* ≤ 0.050) concentrations of α-ketobutyrate and glutamate were observed at the day 10 endpoint relative to control treatment chicks (Figure 8A). At the day 15 endpoint, a higher (*p* ≤ 0.033) concentration of maltose, and lower (*p* ≤ 0.047) concentrations of 2-isopropylmalic acid, glutamate, tryptamine, and tyrosine were observed in virginiamycin relative to control treatment birds not administered the transplant. For the virginiamycin treatment birds administered the microbiota transplant, lower (*p* ≤ 0.034) concentrations of 2-isopropylmalic acid, glucose, and valine were observed at the day 10 endpoint relative to control treatment chicks (Figure 8C). At the day 15 endpoint, a higher (*p* = 0.028) concentration of valine, and lower (*p* ≤ 0.037) concentrations of α-ketobutyrate, glutamate, glucose, and isoleucine were observed in the virginiamycin treatment relative to control treatment birds administered the transplant.

**Figure 7 animals-13-00997-f007:**
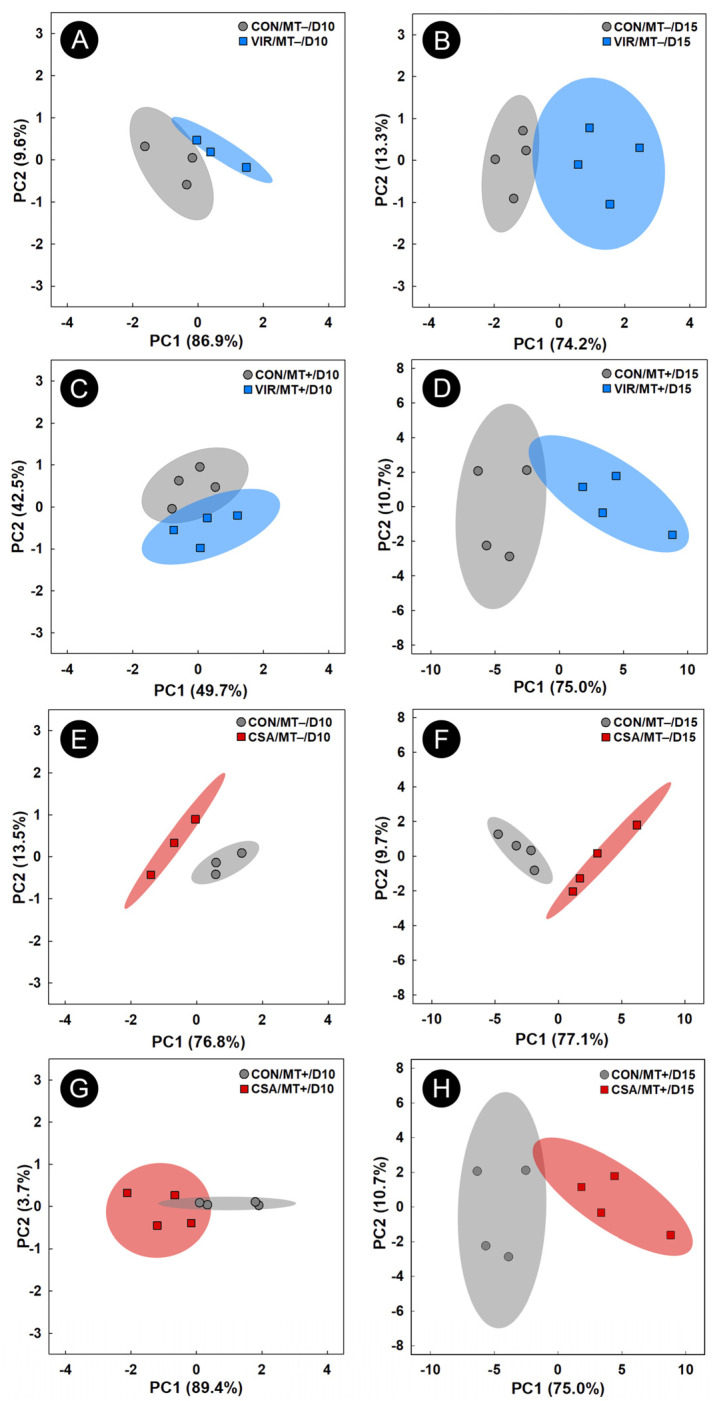
Principal component analysis score plots of metabolite bins in cecal digesta of broiler chicks administered an unamended starter diet (CON), a diet supplemented with virginiamycin (VIR), or a diet supplemented with the ceragenin, CSA-44 (CSA). Birds received a microbiota transplant (MT+) or buffer alone (MT−), and samples were analyzed 10 days post-hatch (D10) and 15 days post-hatch (D15). Each marker represents one bird, and shaded ellipses represent 95% confidence intervals. The x- and y-axes show principal components one and two, respectively, with the number in brackets indicating the percent variance of each component. Metabolite bins were determined to differ (*p* ≤ 0.050) according to a Mann–Whitney U-test. (**A**) CON/MT– vs. VIR/MT– at day 10; (**B**) CON/MT– vs. VIR/MT– at day 15; (**C**) CON/MT+ vs. VIR/MT+ at day 10; (**D**) CON/MT+ vs. VIR/MT+ at day 15; (**E**) CON/MT– vs. CSA/MT− at day 10; (**F**) CON/MT– vs. CSA/MT− at day 15; (**G**); CON/MT+ vs. CSA/MT+ at day 10; (**H**) CON/MT+ vs. CSA/MT+ at day 15.

For CSA treatment birds not administered the microbiota transplant, higher (*p* ≤ 0.044) concentrations of isobutyrate and 2-isopropylmalic acid, and lower (*p* ≤ 0.039) concentrations of glutamate and isoleucine were observed at the day 10 endpoint relative to control treatment chicks (Figure 8B). At the day 15 endpoint, higher (*p* ≤ 0.042) concentrations of isoleucine and valine, and lower (*p* ≤ 0.048) concentrations of glucose-1-phosphate and formate were observed in CSA relative to control treatment birds not administered the transplant. For CSA treatment birds administered the microbiota transplant, a higher (*p* ≤ 0.047) concentration of α-ketobutyrate, and lower (*p* ≤ 0.048) concentrations of glucose and glutamate were observed at the day 10 endpoint relative to control treatment chicks (Figure 8D). At the day 15 endpoint, a higher (*p* = 0.010) concentration of 2-isopropylmalic acid, and lower (*p* ≤ 0.048) concentrations of glucose-1-phosphate and fructose were observed in CSA relative to control treatment birds administered the microbiota transplant.

**Figure 8 animals-13-00997-f008:**
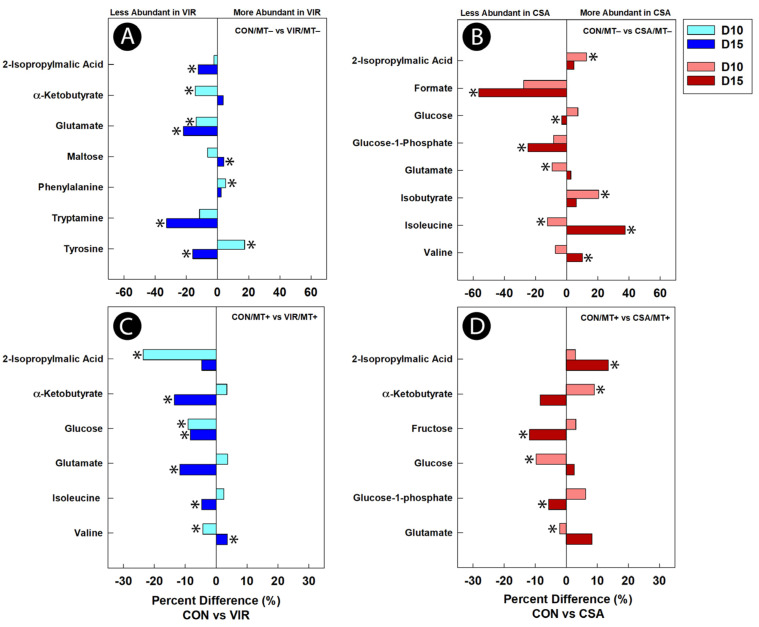
Average percent difference in discriminated metabolites in cecal digesta between broiler chicks that were administered an unamended starter diet (CON), a diet supplemented with virginiamycin (VIR), or a diet supplemented with the ceragenin, CSA-44 (CSA). Birds received a microbiota transplant (MT+) or buffer alone (MT−), and samples were analyzed 10 days post-hatch (D10) and 15 days post-hatch (D15). Metabolites that were altered (*p* ≤ 0.050) according to univariate Mann–Whitney analysis are denoted with an asterisk. (**A**) CON/MT– vs. VIR/MT– at day 10 and day 15; (**B**) CON/MT– vs. CSA/MT− at day 10 and day 15; (**C**) CON/MT+ vs. VIR/MT+ at day 10 and day 15; (**D**) CON/MT+ vs. CSA/MT+ at day 10 and day 15.

## 4. Discussion

### 4.1. Dietary Administration of a Non-Therapeutic Dose of Virginiamycin or Ceragenin CSA-44 Did Not Promote Growth of Broiler Chicks

We investigated the use of the ceragenin, CSA-44 [21] as a novel AGP, along with oral transplantation of cecal microbiota as a method of improving bird growth and increasing enteric bacterial community diversity in young broiler chickens. Additionally, the individual effects of virginiamycin, CSA-44, and microbiota transplantation on the intestinal community establishment, structure, and function were examined. Ceragenins are synthetic amphipathic molecules that are designed to mimic cationic AMPs, and while their distinctive antimicrobial properties have been demonstrated [32], the efficacy of ceragenins as growth promoters is not known. In the current study, we observed that CSA-44 administration alone did not have a significant effect on bird growth over the 15-day duration of the experiment. Similarly, the conventional AGP, virginiamycin, did not impact the growth of the birds. It is noteworthy that the study was not designed to detect the subtle growth promotion effects of AGPs. Rather, the primary aim of the study was to obtain information on the mechanism(s) by which AGPs may promote growth (e.g., via modulation of the structure and function of the enteric microbiota). In this regard, the relatively low number of replicates (*n* = 8), the manner in which birds were maintained (i.e., in IVCs), and the short duration of the study (i.e., 15 days) would have all affected our ability to detect growth enhancements imparted by CSA-44 and virginiamycin. The typical length of broiler chicken production cycles is 35 to 45 days, and previous studies conducted with virginiamycin have shown no significant impact on the body weight of birds at the starter stage but higher body weights at later stages of bird development [33]. Moreover, chicks would not have encountered abnormal immunological stress in the current study, and as a result, the impacts of virginiamycin or CSA-44 as direct or indirect suppressors of inflammation (e.g., incited by pathogens such as *C. perfringens*), including as potential immunomodulatory AGPs would not have been detected (i.e., consistent with the immunomodulatory hypothesis of AGP action [9]).

The establishment of a diverse enteric microbiota is essential for nutrient acquisition and immune development [1], and the intestinal microbiota confers direct and indirect protection against intestinal pathogens [2]. One mechanism by which the microbiota confers a health benefit is via the production of SCFAs, especially butyrate, which has been shown to confer protection against *Salmonella enterica* and *Clostridium perfringens* [34,35]. In the current study, we evaluated SCFA production by quantifying their concentration in feces 5, 10, and 15 days post-hatch. However, with the exception of acetic acid, concentrations of SCFAs were present below the limit of detection. A previous study conducted on young birds infected with *S. enterica* serovar Typhimurium showed similar results [36]. The low concentration of these components could be associated with sampling time and intestinal transit. In this regard, reduced quantities of digesta will move from the ileum into the cecum for fermentation, with the majority of digesta directly voided into the colon and eliminated with the feces [37,38]. Thus, products of fermentation found in chicken feces will be influenced by whether sampling is conducted at the moment of cecal excretion [38], and the low concentrations of SCFA that we observed could be the result of sampling feces that only contained ileal digesta.

### 4.2. Microbiota Transplantation Impacted Cecal Bacterial Community Composition and Diversity

A diverse cecal microbiota has been shown to improve bird growth and performance [39]. Additionally, variations in taxa abundance and α- and β-diversity have been shown to be directly associated with growth differences among birds [40]. Previous studies by Inglis et al. [26] and Zaytsoff et al. [2] have shown that oral administration of a microbiota transplant increased the diversity of cecal microbiota and established a similar enteric bacterial community structure within birds housed in separate cages. In the current study, we questioned if the administration of a complex cecal microbiota could establish a bacterial community that would be stable over time and minimize performance variation among birds. We observed that the administration of the microbiota transplant significantly increased the bacterial diversity in the cecum, resulting in a consistent community structure within birds and across all treatment groups. Neither α- or β-diversity of bacterial communities in cecal digesta of birds administered the microbiota transplant differed between the day 10 and day 15 endpoints, suggesting that the microbiota established and stabilized earlier than day 10 and remained unchanged thereafter. Conversely, animals that were not administered the microbiota transplant showed α- and β-diversity differences in bacterial communities between the 10- and 15-day endpoints. Previously conducted studies have attempted to use microbiota transplantation (e.g., via administration to egg surfaces) to improve growth and feed-efficiency profiles of broiler chickens with limited success, but did observe reduced variation in the cecal microbiota among birds [5,40,41]. In this regard, Stanley et al. [3] described a high variability of cecal microbiota diversity among birds that were not subjected to a microbiota transplant in three separate trials. In our experiment, birds were maintained in IVCs connected to a high-efficiency particulate air filter unit for the duration of the experiment. Thus, the introduction of bacteria from the local environment was negligible in the current study, which likely contributed to the low degree of inter-bird variability that we observed. It is noteworthy that we observed a high degree of inter-bird variability in the structure of bacterial communities in chicks that were not administered the microbiota transplant. Using the same experimental system, we previously demonstrated that birds orally administered a microbiota transplant were more resistant to necrotic enteritis following challenge with *C. perfringens*, which was attributed to increased α-diversity and altered β-diversity in the cecum and small intestine [2]. Increasing bacterial diversity with minimal inter-bird variability through microbiota transplantation may prove to be a beneficial strategy for researchers, and ultimately, the chicken industry.

It is noteworthy that broiler chicks administered the microbiota transplant exhibited comparable bacterial diversity to that of the cecum of healthy adult donors (Shannon’s α diversity index of ≈6.5) [42], including bacteria within important SCFA-producing families such as *Lachnospiraceae* and *Ruminococcaceae* [43,44]. Moreover, chicks that were administered the transplant possessed a significantly higher abundance of bacteria within the families *Butyricicoccaceae*, *Acidaminococcaceae*, *Sutterellaceae*, *Peptococcaceae*, and *Atopobiaceae* as compared to chicks not administered the microbiota transplant. All of these families have been directly associated with the production of SCFAs, including butyrate.

### 4.3. Microbiota Transplantation Impacted Cecal Bacterial Function

To evaluate the impacts of AGP and microbiota transplant treatments on the function of the microbiota, we conducted a metabolomic analysis of cecal digesta. Metabolomics is a powerful tool that can be used to examine and characterize the function of microbiota by quantifying metabolites and examining alterations in biochemical pathways [31]. Enteric microorganisms within the intestinal tract of broiler chickens are involved in numerous host functions, such as digestion, the production of beneficial dietary metabolites, and the regulation of metabolism [32]. We observed the metabolome differed in birds administered the microbiota transplant. In this regard, we observed higher concentrations of isoleucine, 2-isopropylmalic acid, and alanine within control treatment birds at the day 10 endpoint. At the day 15 endpoint, we observed an increase in glucose-6-phosphate, and decreases in glucose and glucose-1-phosphate. Glycolysis begins with the breakdown of glucose into glucose-6-phosphate and ends with the production of pyruvate after a series of reactions. Pyruvate participates in many biochemical pathways, including the synthesis of the essential branched-chain amino acid (BCAA), isoleucine [45]. Isoleucine can be synthesized by bacteria through several reactions beginning with the deamination of threonine into α-ketobutyrate. Pyruvate is then utilized to proceed further with reactions that result in the synthesis of isoleucine [46]. Pyruvate is also a direct precursor for the synthesis of alanine [47]. 2-Isopropylmalic acid is a direct intermediate in the synthesis of leucine and is derived from valine synthesis or via pyruvate metabolism [45]. While glucose and glucose-6-phosphate were significantly altered at the day 15 endpoint, alterations in the concentrations of these metabolites along with isoleucine, 2-isopropylmalic acid, and alanine suggest that the administration of the microbiota transplant resulted in an upregulation of glycolysis, and BCAA synthesis was therefore upregulated within the cecal digesta metabolome. Glucose-1-phosphate is a direct product of the breakdown of glycogen, a primary molecule for the storage of energy in animals and bacteria [48]. Glucose-6-phosphate can also be converted to glucose-1-phosphate to participate in the formation of glycogen when carbohydrates are in excess or when bacteria are at a stationary growth phase [49]. Glycogen is not required for bacterial growth but tends to accumulate in these conditions [49]. Therefore, the lower concentration of glucose-1-phosphate that was observed in birds administered the microbiota transplant could indicate that the cecal microbiota was utilizing glucose more efficiently as a result of the transplant, and this could contribute to the enhancement of bird health by increasing essential BCAAs.

### 4.4. Microbiota Transplantation Temporally Impacted the Cecal Digesta Metabolome

A comparison of the two endpoints for control treatment birds not administered the microbiota transplant (i.e., in which α- and β-diversity of bacteria increased over time) revealed a higher concentration of isoleucine at the day 15 endpoint. As isoleucine is an essential BCAA for broiler chickens, an increase in the concentration of this amino acid could be directly associated with the increase in bacterial diversity observed, which could be beneficial for bird health. Concentrations of 2-isopropylmalic acid, taurine, and tyrosine were downregulated at the day 15 endpoint relative to the day 10 endpoint in chicks not administered the microbiota transplant. While taurine is not considered to be an essential amino acid and can be synthesized de novo by broiler chickens [50], it has been demonstrated to have antioxidant and anti-inflammatory properties and to reduce heat stress in poultry [51,52]. Additionally, dietary supplementation with taurine has been shown to increase the expression of appetite genes and increase bird weight in heat stress conditions [53,54]. Consistent with the increase in diversity of the cecal microbiota observed at the day 15 endpoint, it is possible that taurine production was upregulated to mitigate inflammation caused during early bacterial community establishment [2]. Consistent with this possibility, taurine decreased once the microbiota community structure stabilized. Tyrosine is one of the three essential aromatic amino acids (AAAs), together with phenylalanine and tryptophan, that chickens can not synthesize [55]; therefore, these AAAs need to be obtained through their diet or via *de novo* synthesis by enteric bacteria via the shikimate pathway [56]. The observed increase in isoleucine, coupled with the decrease in tyrosine concentrations at the day 15 endpoint, could be attributed to the cecal microbiota utilizing carbohydrates for other amino acid synthesis processes, such as the synthesis of isoleucine, which could be supported by the absence of other metabolites involved in the shikimate pathway.

A temporal examination of the metabolome in birds administered the microbiota transplant revealed higher concentrations of glutamate, isoleucine, and taurine at the day 15 endpoint. Glutamate is not only a proteinogenic amino acid, but it is also important as an amino group donor for nearly all metabolites containing nitrogen [57]. It is synthesized from α-ketogluturate, a tricarboxylic acid (TCA) cycle intermediate, and it is a highly abundant metabolite in bacterial cells in all nutrient conditions [57]. *Acidaminococcaceae* and *Peptococcaceae* are families of bacteria that contain species able to ferment glutamate into the SCFAs, acetate and butyrate [58,59]. Therefore, the increase in the concentration of glutamate observed within control treatment birds administered the microbiota transplant may not only be the result of the increased bacterial diversity, but may also contribute to the production of SCFAs later in the broiler chicken life cycle, as the related SCFA-producing families were present in the birds administered the transplant. The observed increase in isoleucine over time supports the possibility that the increased diversity of bacteria in birds administered the transplant is beneficial for increased production of BCAAs. As described previously, taurine has been demonstrated to function as an antioxidant and anti-inflammatory agent that can be synthesized *de novo* by broiler chickens [52,53,54]. Thus, the increase in taurine concentrations could be influenced by the increase in bacterial diversity resulting from microbiota transplantation.

We observed that glucose, glucose-1-phosphate, and leucine concentrations were decreased in birds administered the microbiota transplant at the day 15 endpoint relative to the day 10 endpoint. The observed decrease in glucose could be attributed to the high diversity of bacteria achieved through transplantation and the ensuing metabolic activity of the microbiota. In this regard, the decrease in glucose and glucose-1-phosphate is consistent with increased utilization of carbohydrates and comparatively reduced storage of glycogen, and the increased production of amino acids, such as isoleucine. The level of synthesis of BCAAs is dependent on the availability of pyruvate [45], which could explain why leucine was decreased while isoleucine was increased at the day 15 endpoint. Overall, the substantive upregulation of BCAA pathways following microbiota transplantation may be an important mechanism conferring a potential health benefit to broiler chickens.

### 4.5. Dietary Administration of Virginiamycin Affected the Cecal Digesta Metabolome

Many antibiotics are known to target energy-consumption pathways, and therefore, bacterial metabolism [60]. Virginiamycin is a streptogramin antibiotic that is known to target bacterial protein synthesis in Gram-positive bacteria by inhibiting aminoacyl-tRNA incorporation into the 50S ribosomal subunit and inhibiting translation of mRNA [61,62]. At non-therapeutic concentrations, virginiamycin has been utilized in commercial production to beneficially modulate the intestinal microbiota, thereby enhancing bird growth. Neumann et al. [62] observed that virginiamycin administered at a rate of 20 mg/kg in feed resulted in an overall decrease in bacterial diversity in the cecum of mature broilers and an enrichment of bacteria within the *Oscillospiraceae* family, which contains butyrate-producing species such as *Faecalibacterium prausnitzii*. They concluded that the increase in abundance of this butyrate-producing family might be responsible for the increase in feed efficiency observed. In the current study, bacteria within the family *Oscillospiraceae* were present, but there was no difference in their abundance between control birds and those administered the microbiota transplant. Moreover, the administration of virginiamycin, with or without the microbiota transplant, did not increase their relative abundance. Another study by Chen et al. [63] showed that virginiamycin administered at a rate of 30 mg/kg in feed increased the relative abundance of *Bacteroides*, *Lachnospiraceae*, and members of *Enterobacteriaceae* in the cecum. While we did observe the presence of these bacterial families across treatments, we did not observe a difference in their abundance. The lack of significant taxa differences could be attributed to the concentration of virginiamycin that we administered (20 mg/kg in feed), which is a lower concentration than that used by Chen et al. [63]. However, Neumann et al. [62] observed taxonomic changes using a dose of 20 mg/kg of virginiamycin in feed, and it is likely that the highly controlled environment in which we reared birds reduced the introduction of bacteria from the local environment. While the structure of the cecal bacteria was not altered by virginiamycin at the non-therapeutic dose examined in the current study, the antibiotic may have inhibited protein-synthesis pathways in certain bacterial taxa, thereby reducing competition and favoring SCFA-producing taxa.

We further employed metabolomics to ascertain the impact of virginiamycin on the bacterial function within the cecal digesta. To our knowledge, the study performed by Chen et al. [63] is the only other study conducted to date that has reported cecal metabolome responses to virginiamycin in broiler chickens; in contrast to us, they used LC-MS-based metabolomics. Significant differences in the metabolome of birds administered virginiamycin without the microbiota transplant relative to control treatment chicks (i.e., also not administered the transplant) were observed at both experimental endpoints in the current study. At the day 10 endpoint, we observed higher concentrations of phenylalanine and tyrosine, and lower concentrations of α-ketobutyrate and glutamate in birds administered virginiamycin without the microbiota transplant relative to control treatment birds not administered the transplant. The increase in phenylalanine and tyrosine concentrations is suggestive of an upregulated shikimate pathway [56], and the decrease in α-ketobutyrate may result in reduced production of isoleucine. It is noteworthy that Chen et al. [63] also observed that virginiamycin affected the phenylalanine pathway. Additionally, the decrease in glutamate that we observed could reduce the production of SCFAs, including acetate and butyrate by bacterial families such as *Acidaminococcaceae* and *Peptococcaceae* [58,59]. At the day 15 endpoint, higher concentrations of maltose and lower concentrations of 2-isopropylmalic acid, glutamate, tryptamine, and tyrosine were observed in birds administered virginiamycin alone without the microbiota transplant relative to control treatment birds not receiving the transplant. While there was an increase in the AAAs, phenylalanine and tyrosine at the day 10 endpoint, we observed a decrease in tyrosine at the day 15 endpoint. We also observed a significant decrease in 2-isopropylmalic acid, the precursor to the essential BCAA leucine. The observed reduction in glutamate observed at the day 15 endpoint is consistent with a bacteriostatic effect of virginiamycin. Tryptamine is a metabolite known to be produced by intestinal bacteria as a derivative of the AAA, tryptophan [64]. Thus, the reduced levels of tryptamine and tyrosine could be an indication that the shikimate pathway was downregulated at the latter time, an effect that could be attributed to the virginiamycin treatment on the more diverse microbiota. The diversity of bacteria in birds administered virginiamycin without the microbiota transplant versus control treatment chicks not provided the transplant was equivalent, suggesting that virginiamycin effects on shikimate pathway metabolites and metabolites related to BCAA synthesis during bacterial community establishment are an underlying mechanism of action of the AGP that is independent of potential impacts of the AGP on the enteric bacterial community structure.

Significant differences in the metabolome of birds administered a microbiota transplant and virginiamycin relative to control treatment chicks administered the transplant were observed at both experimental endpoints. At the day 10 endpoint, lower concentrations of 2-isopropylmalic acid, glucose, and valine were observed in birds administered virginiamycin and the microbiota transplant relative to control birds administered the transplant. The decrease in 2-isopropylmalic acid and valine is another instance of altered BCAA synthesis pathways due to the virginiamycin treatment. The observed decrease in glucose could also contribute to the downregulation of these pathways via reduced glycolysis. At the day 15 endpoint, a higher concentration of valine, and lower concentrations of α-ketobutyrate, glutamate, glucose, and isoleucine were observed. The increase in the concentration of valine differed at the day 10 endpoint, whereas the reduction in glucose was consistent at both sample times. It is possible that the reduction in glucose concentrations resulted in diminished production of pyruvate, thereby downregulating the synthesis of other BCAAs, such as isoleucine. The consistent alteration of BCAA synthesis pathways in the virginiamycin-treated birds was conspicuous, especially at the day 15 endpoint, where the bacterial diversity was increased in all birds administered the microbiota transplant. Further research to determine if the alterations in BCAA and AAA synthesis by virginiamycin can be replicated, and whether this occurs at different locations in the gastrointestinal tract of broiler chickens is warranted.

### 4.6. Dietary Administration of Ceragenin CSA-44 Affected the Cecal Digesta Metabolome

The use of cathelicidin mimics as antimicrobial agents is an emerging area of investigation. Their bacteriostatic and bactericidal effects have primarily been studied in vitro, and in vivo evaluation and the elucidation of the mechanisms of action have received limited examination to date. Previous studies have demonstrated that ceragenins are effective against Gram-positive and Gram-negative bacteria, as well as mycobacterial species [32]. CSA-44, the molecule used in the current study is bactericidal to a variety of bacteria, including *Escherichia coli*, *Enterobacter cloacae*, and *Klebsiella pneumonia* [65]. Additionally, a recent study by Tokajuk et al. [21] observed that CSA-44 was able to prevent and reduce biofilm formation by oral bacterial and fungal pathogens, including *Enterococcus faecalis* and *Candida albicans*. This suggests that ceragenins may have value as AGPs. Similarly to virginiamycin, we observed that the administration of CSA-44 with or without the microbiota transplant did not alter the structure of the cecal microbiota, but did result in significant alterations to the metabolome of cecal digesta. At the day 10 endpoint, higher concentrations of 2-isopropylmalic acid and isobutyrate, and lower concentrations of glutamate and isoleucine were observed in birds administered CSA-44 without the microbiota transplant relative to control treatment birds also not administered the transplant. The increased concentration of 2-isopropylmalic acid is indicative of upregulated leucine synthesis [46]. Isobutyrate is a precursor for the SCFA, butyrate [66]. As indicated previously, the downregulation of glutamate can also affect the production of other SCFAs, including acetate and butyrate [58,59]. At the day 15 endpoint, higher concentrations of isoleucine and valine, and lower concentrations of formate, glucose, and glucose-1-phosphate were observed in birds administered CSA-44 without the microbiota transplant relative to control treatment birds also not administered the transplant. Higher utilization of glucose is consistent with a higher rate of glycolysis, along with decreased production of glucose-1-phosphate. Pyruvate formed from glycolysis can then be used as the major precursor to produce the essential BCAAs, isoleucine and valine. The observed downregulation of isoleucine at the day 10 endpoint, and upregulation of isoleucine at the day 15 endpoint can be explained by the higher diversity of bacteria at the latter endpoint. Formate is a SCFA that has been known to limit pathogens, including *Salmonella* spp., and it has been examined as a poultry diet amendment [67]. Formate is also an important metabolite in the energy metabolism of *E. coli* and other enterobacteria [68]. Although CSA-44 administration did not affect bacterial diversity, the observed reduction in formate concentration may serve as a marker for the antimicrobial activity of CSA on formate-producing bacteria, such as those within the family *Oscillospiraceae* [69].

Significant differences in the metabolome of birds administered the microbiota transplant and CSA-44 relative to control treatment chicks administered the transplant were observed at both experimental endpoints. At the day 10 endpoint, higher concentrations of α-ketobutyrate, and lower concentrations of glucose and glutamate were observed in birds administered CSA-44 and the microbiota transplant relative to control birds also administered the microbiota transplant. α-Ketobutyrate can be utilized by bacteria for the production of isoleucine [46], and glucose may be consumed at a higher rate to assist in these pathways. The decrease in glutamate concentrations could be attributed to an antimicrobial effect of CSA-44 or to the decrease in glucose. A decrease in glucose could downregulate glycolysis and TCA cycle products, including glutamate, via reduced pyruvate synthesis. At the day 15 endpoint, higher concentrations of 2-isopropylmalic acid, and lower concentrations of fructose and glucose-1-phosphate were observed in birds administered CSA-44 and the microbiota transplant relative to control birds also administered the microbiota transplant. An increased concentration of 2-isopropylmalic acid is indicative of the synthesis of the essential BCAA leucine [45], and lower concentrations of fructose and glucose-1-phosphate are consistent with the higher utilization of carbohydrates and decreased glycogen formation. In broiler chicks administered CSA-44, we primarily observed alterations in glycolysis and BCAA synthesis pathways, and in contrast to virginiamycin treatment, we did not observe any changes in key metabolites involved in the shikimate pathway and AAA synthesis. The substantial effect that CSA-44 had on the metabolome of birds in the current study indicates that the elucidation of the mechanisms and impacts of ceragenins on broiler chicken health throughout the production period is warranted.

## 5. Conclusions

We showed that the oral administration of a microbiota transplant obtained from a healthy adult broiler donor to neonatal broiler chicks resulted in the establishment of a highly diverse and homogeneous enteric bacterial community across birds. The transplant was also shown to increase the abundance of SCFA-producing bacterial families. Neither the administration of virginiamycin nor ceragenin CSA-44 increased the growth performance of birds over the 15-day duration of the experiment. Moreover, neither agent altered the structure of bacterial communities. However, characterization of the metabolome using ^1^H-NMR spectroscopy demonstrated that both virginiamycin and CSA-44, as well as the microbiota transplant, altered the metabolome of cecal digesta, including energy and amino acid synthesis pathways. Thus, AGPs and microbiota transplantation affected the function but not the structure of enteric bacteria. Future research should examine the impacts of AGPs, including ceragenins, for a protracted period and in simulated production settings, and also elucidate the degree to which AGPs alter energy, BCAA, and AAA pathways in relation to metrics of chicken health, including in birds under immunological stress.

## Data Availability

The microbiota raw sequencing reads were submitted to the Sequencing Read Archive of NCBI under BioProject accession number PRJNA927030.

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
