# Peer review of "Antimicrobial Growth Promoters Altered the Function but Not the Structure of Enteric Bacterial Communities in Broiler Chicks ± Microbiota Transplantation"

_animals, 2023, doi:10.3390/ani13060997_

Round 1
Reviewer 1 Report
Please see the attachment.

Author Response
Reviewer #1
Author general comment. We appreciate the comprehensive review and constructive comments provided by the reviewer to improve the manuscript.
Specific comment
1. Reviewer comment. It s a 3 x2 design. However, the aims of this study are not descripted clearly. What were the reasons to combine cathelicidin mimic, CSA-44, and virginiamycin treatments and cecal microbiota transplantation. The combination made the manuscript is hardly to follow.
Author response. The study aimed to investigate the effects of a conventional antimicrobial growth promoter (AGP) (Virginiamycin) and an alternative AGP candidate (CSA-44) on the cecal microbiota and cecal metabolome. The use of host-defense peptides (HDPs), including antimicrobial peptide (AMP) mimics is an active area of research in human medicine, and we chose to examine the impact of a novel cathelicidin mimic as a potential growth promoter. We have shown that early establishment of a diverse microbiota in chicks has profound impacts on the health of the birds, and the microbiota transplant variable was included to address the impact of the early establishment of a diverse and homogeneous microbiota on the AGP-host-microbiota interaction. Importantly, a primary purpose of the study was to obtain information on the potential mechanisms by which AGPs may provide a health benefit to broilers. We have added additional information to the introduction section to better define the aims of the study. In addition, we have added a supplemental figure describing the experimental design.
2. Reviewer comment. It is not clear about the sample size, and it is not mentioned at any place. How did get 4 birds/treatment (Line 120) and two runs with two replicates/each run (48 animals total) (Line 121). Based on the current writing: It was 4 birds/time point/treatment x 2 time point/treatment x 3 treatments/level x 2 levels/run x 2 runs = 96 birds. What was the sample size for statistical analysis, four birds (replicates)/treatment or two replicates/run?
Author response. We have added a supplemental figure to provide the reader with a pictorial description of the experimental design employed. The larger experiment was a two (microbiota transplant; MT) by three (antimicrobial growth promoter; AGP) by two (endpoint; E) factorial experiment arranged as a completely randomized design with four replicates. However, in light of the concern expressed by the reviewer regarding the small number of replicates for the growth promotion metrics, we have re-analyzed the data with eight replicate chicks (i.e., we removed the endpoint variable and analyzing data averaged over days). Following is the degrees of freedom table for the revised growth promotion analyses.
|
Source |
df |
|
AGP (A) |
2 |
|
MT (M) |
1 |
|
A*M |
2 |
|
Error |
42 |
|
Total |
47 |
The four replicates were conducted in each of two runs (eight total). “Run” was treated as a random effect in the statistical model.
3. Reviewer comment. Need much detail information about the transportation, such as that how long it took from the hatchery to the research facility; the containers were used; and how to avoid the environmental exposure affects birds gut microbiota composition?
Author response. We have added additional information on the transport of the chicks from the hatchery. In this regard, the hatchery is located 5 minutes away from the Lethbridge Research and Development Centre. Animals were transported in a container fully wrapped with autoclaved surgical drapes, to guarantee the sterility of the container. Since all the animals were subjected to the same conditions any impact that the environment could have had in the microbiota would be equal across animals and treatments.
4. Reviewer comment. It is not clear: How the “chicks were immediately placed in individually ventilated cages” (Line 126) became “two replicate chicks were housed together per cage (Line 134). Were the 2 birds belonged to the same treatment but different runs?
Author response. This has been clarified in the revised manuscript. Immediately upon arrival at Lethbridge Research and Development Centre, the sterile transport container was placed in an operating biosafety cabinet (BSC), the transport drapes were removed, and chicks were individually weighed. Chicks were then randomly transferred into a sterile individual ventilated cage (IVC) within the BSC. Cages were ready before the arrival of the birds at the animal facility. Two randomly selected chicks were placed in each IVC as is required by our Animal Care Committee. Although housed together the two chicks were assigned to the same treatment, but they were treated independently (e.g., with regard to microbiota transplantation). As soon as chicks were placed in the cages, the cages were transferred to the Techniplast cage rack operated in containment mode.
5. Reviewer comment. “… from healthy adult broilers. (Line 137)”. How many adult broilers were used? How were the adult birds housed; and how old of them?
Author response. Adult birds were obtained from a local farm from birds of known health status (six birds). The housing of these animals corresponded with production conditions of Ross 308FF based on the Aviagen Ross Broiler Management Handbook (2018). Donor chickens were 25 weeks-of-age.
6. Reviewer comment. “the slurry was dispensed ..” (Line 150). Individual or mixed donor cecal samples were used; and what was the concentration of cecal microbiota used?
Author response. The slurry was generated by combining equal amounts of cecal digesta from all six donor birds. The slurry was prepared by mixing 25% combined digesta and 75% reduced glycerol + Columbia broth (v/v). Thus, in the 200 µL of inoculum, 50 µL corresponded to cecal digesta. Additional information has been added to the manuscript.
7. Reviewer comment. “on the day of the transplantation, … (Line 152). Need to give at what age (how old of the birds) the treatment was applied. Need to give the reasons why the treatment was not conducted at 1 day-old chicks and discuss how the rearing environment may affect developing of gut microbiota composition during the 1st 2 – 3 days.
Author response. Chickens were orally transplanted with the intestinal microbiota on day 1 of the experiment (1-day-old chicks). This information has been added to the manuscript, including the addition of a supplemental figure pictorially describing the experiment design. We did not discuss how rearing environment may affect the developing of the microbiota since this has been previously described by Kers et al. (2019). This reference has been added to the manuscript.
8. Reviewer comment. “On day 3, …..” (Lines 161-164). Need to give all the treatments based on the 3 x 2 design.
Author response. A supplemental figure pictorial describing the treatments has been added.
9. Reviewer comment. Were the fresh feces collected from 2 birds/cage (Line 167)? Not clear the meaning of Individual treatments here (line 168).
Author response. The sentence indicating “individual treatments” has been removed.
10. Reviewer comment. What was the statistical analysis unit, and the sample size per treatment?
Author response. See response to Comment #2 above (e.g., source and degrees of freedom table). For the larger experiment, there were four replicates per treatment (see new supplemental figure pictorially describing the experimental design). As indicated above, for the analysis of growth promotion, we re-analyzed the data with eight replicate chicks (i.e., we removed the endpoint variable and analyzing data averaged over days).
11. Reviewer comment. Line 57. “gender, age, breed” are belonged to the internal factors
Author response. “Intrinsic factors” has been added to the sentence.
12. Reviewer comment. Line 60. Change to “thereby affect bird nutrition and response to various treatments”
Author response. The sentence has been modified as suggested.
13. Reviewer comment. Line 65. Change “that” to “of” and delete “has”
Author response. The sentence has been modified as suggested.
14. Reviewer comment. Line 77. Change to “result of”
Author response. The sentence has been modified as suggested.
15. Reviewer comment. Lines 91 - 92. Delete “in broiler chickens by”, change “reducing” to “reduce”, and add “in broiler chickens”
Author response. The sentence has been modified.
16. Reviewer comment. Line 92. Change to “have been specially focused”
Author response. The sentence has been modified to enhance clarity.
17. Reviewer comment. Line 99. Change to “Ceragenins, as non-peptide mimics of cathelicidins, are …”
Author response. The sentence has been modified as suggested.
18. Reviewer comment. Line 100. Delete “they”
Author response. The sentence has been modified as suggested.
19. Reviewer comment. Line 104. What is CSA-44? It needs to give above.
Author response. Additional information has been added justifying the use of the host-defense peptide mimic, CSA-44. See our response to Comment #1 above.
20. Reviewer comment. Line 105. Change “enhancing” to “to enhance”
Author response. The sentence has been modified as suggested.
21. Reviewer comment. Lines 290-292. Rewrite the sentence
Author response. The sentence has been modified.
Reviewer 2 Report
The manuscript of Hodak et al. reported effects of oral administration of antimicrobial compounds and microbiota transplantation on cecal microbiota composition and functions in broilers. The topic of revealing the mode of action of growth promoting antibiotics and probiotics is interesting to readers. This manuscript is well written and fits in this journal. However, the manuscript needs to be revised before publishing.
Major comments,
Experimental design was not ideal to fulfill authors hypothesis. The replicate numbers for broilers and cages were not enough to draw convincible conclusions, especially the overall design was based off 2 repeat trials. Experimental design contained too many factors, which led to 12 potential comparisons among 3 levels of dietary treatments, 2 levels of transplantation, and 2 levels of sampling time points. The rationale of selected comparison in the result seems not supported by hypothesis. The use of microbe-controlled individually ventilated cages was not ideal for investigating the oral antimicrobial compounds and microbiota transplantation, thus limited the practical meanings of the manuscript.
Minor comments,
Line 48-114. Introduction. Need information regarding the rationale of microbiota transplantation and the potential synergistic effects with antimicrobial compounds.
Line 116-121. Add rationale of choosing d10 and d15 at sampling time points. How was the number of replicate (n=2) determined, especially the overall design was repeated, which resulted in more variations.
Line 130-131. The ambient temperature is colder than optimal. Please explain.
Line 242-255. Result analysis for growth performance is not clear. Suggest adding p values for main effects and their interactions on the figure. What is experimental unit (EU) for growth performance? replicate number is needed on the figure. if one cage is considered as EU, it seems n=2, which is not a reasonable replicate. The feed conversion ratio data is missing from growth performance, please add the data to results.
Line 332 -395. Why only focus on the control group but do not compare microbiota transplant effect under antimicrobial compound?
Author Response
Reviewer #2
General comment
1. Reviewer comment. The manuscript of Hodak et al. reported effects of oral administration of antimicrobial compounds and microbiota transplantation on cecal microbiota composition and functions in broilers. The topic of revealing the mode of action of growth promoting antibiotics and probiotics is interesting to readers. This manuscript is well written and fits in this journal. However, the manuscript needs to be revised before publishing.
Author response. The reviewer has identified points that required clarification/ justification, which have all been addressed within the manuscript and/or in our responses to the specific comments. We thank the reviewer for the time that they expended on reviewing the manuscript.
Specific comments
1. Reviewer comment. Experimental design was not ideal to fulfill authors hypothesis. The replicate numbers for broilers and cages were not enough to draw convincible conclusions, especially the overall design was based off 2 repeat trials. Experimental design contained too many factors, which led to 12 potential comparisons among 3 levels of dietary treatments, 2 levels of transplantation, and 2 levels of sampling time points. The rationale of selected comparison in the result seems not supported by hypothesis. The use of microbe-controlled individually ventilated cages was not ideal for investigating the oral antimicrobial compounds and microbiota transplantation, thus limited the practical meanings of the manuscript.
Author response. We respect the reviewer’s comment regarding the number of birds used in this experiment. It was not possible to conduct power analyses as the level of variation to be encountered was unknown. In light of the concern expressed by the reviewer regarding the complexity of the experimental design, we have re-analyzed the data with eight replicate chicks (i.e., we removed the endpoint variable and analyzing data averaged over days). Following is the degrees of freedom table for the revised growth promotion analyses.
|
Source |
df |
|
AGP (A) |
2 |
|
MT (M) |
1 |
|
A*M |
2 |
|
Error |
42 |
|
Total |
47 |
We are unable to humanely maintain broiler chickens in individually ventilated cages (IVCs) beyond 15 days due to their size. A salient goal of the study was to ascertain potential mechanisms by which antimicrobial growth promoters (AGPs) may provide a health benefit to broilers, and the study was not meant to mimic a broiler barn. Importantly, the study was designed to provide information that would inform experiments in simulated and actual production settings. We chose to use IVCs to eliminate external factor that could/would confound our experimental variables. For example, it is well known that the local environment, human contact, water, etc. can impact the establishment and composition of the enteric microbiota; we chose to use IVCs operated in containment mode to remove these influences on the microbiota. Moreover, we utilized IVCs to ensure that birds assigned to individual treatments were maintained under the same conditions.
Despite the limitations of the study to ascertain growth promotion, we feel that the study contains important novel information that should be available to the scientific community. We have added additional text to the discussion section describing the limitations of the study.
2. Reviewer comment. Line 48-114. Introduction. Need information regarding the rationale of microbiota transplantation and the potential synergistic effects with antimicrobial compounds.
Author response. Additional information providing the rationale for the study has been added to the introduction. We have also added additional text to the discussion section describing the limitations of the study.
3. Reviewer comment. Line 116-121. Add rationale of choosing d10 and d15 at sampling time points. How was the number of replicate (n=2) determined, especially the overall design was repeated, which resulted in more variations.
Author response. Since birds were housed in IVCs, the longest that broilers can be maintain in these cages is 15 days. After 15 days broiler chickens outgrow the cages, and due to welfare conditions, they cannot be kept for longer time in the IVCs. Since a primary objective of our study was to evaluate the effect of AGPs on the function of a uniform and homogeneous microbiota, reducing the confounding effects of external factors on the microbiota was of paramount importance. Thus, we chose to use IVCs instead of open cages. We chose the day 10 endpoint to examine the microbiota in chicks at an early stage of establishment. See our description of the re-analysis of the growth promotion metrics using eight replicates instead of four replicate birds per treatment (i.e., see response to Comment 1 above).
4. Reviewer comment. Line 130-131. The ambient temperature is colder than optimal. Please explain.
Author response. We followed the temperature regimen specified in the Aviagen Ross Broiler Management Handbook (2018) (Table 2.2 below). The ambient humidity levels used are mandated by the Canadian Council on Animal Care (CCAC guidelines on: The care and use of farm animals in research, teaching and testing). At no point did we observe any behavior of temperature stress (e.g., abnormal behavior such as huddling).

5. Reviewer comment. Line 242-255. Result analysis for growth performance is not clear. Suggest adding p values for main effects and their interactions on the figure. What is experimental unit (EU) for growth performance? replicate number is needed on the figure. if one cage is considered as EU, it seems n=2, which is not a reasonable replicate. The feed conversion ratio data is missing from growth performance, please add the data to results.
Author response. The information has been added to the Figure S1 (now Figure S3 in the revised manuscript) and Figure S2 as suggested. We treated birds as the experimental unit (i.e., not cages). See response to Specific Comment #1 above for additional information including on the re-analysis of the data, and our statements added to the discussion section on the limitations of our study. Feed conversation ratio results have been added to the manuscript, including new figure, Figure S4.
6. Reviewer comment. Line 332 -395. Why only focus on the control group but do not compare microbiota transplant effect under antimicrobial compound?
Author response. When evaluating specific changes incited exclusively by the administration of the microbiota transplant (MT), we decided to focus on examining individual AGP treatments relative to the control treatment. The manuscript is already quite long (i.e., 22 pages), and we made a decision to not present ± MT and virginiamycin versus ± MT and CSA-44 as it added significant information/text that detracted from the primary take home messages.
Reviewer 3 Report
This study characterized the impact of the antibiotic virginiamycin, and an AGP alternative (non-peptide mimics of cathelicidins): ceragenins (CSA-44), on the structure and function of the broiler chicken cecal microbiota. The authors have used next generation sequencing and 1H-Nuclear Magnetic Resonance Spectroscopy (NMR)-based metabolomics and bioinformatics analysis to support their conclusions. This is a very interesting topic and worth of investigation. However, still, I think this manuscript needs a major revision. The experimental design is not acceptable because the number of replicates is too low to derive conclusions from this study: "two replicates in each run (2 runs), 48 animals in total". Results are well explained, and tools utilized for figures and bioinformatics analyses are accurate. My major concern is weak experimental design for the bird trial. Statistical analyses of feed consumption and bird growth average over time is not likely accurate in a trial of 48 birds. What is statistical power of your bird trial? Experimental design i.e., a trial with 48 birds is not feasible to establish a statistical model. Sample size is too small to achieve statistical significance. I suggest increasing number of birds in each treatment.
Also, I think AGP at low levels are known to increase body weight, however in Line 37 authors mentioned that neither virginiamycin nor CSA-44 influenced weight gain or feed consumption". Please comment.
Any data on use of ceragenins in animal studies? Please comment if its toxicity, and other activities have been evaluated in living organisms? If yes, what levels were effective. Add more information on ceragenins.
Line 44-45: Need more evidence to support this conclusion.
Author Response
Reviewer #3
General comment
1. Reviewer comment. This study characterized the impact of the antibiotic virginiamycin, and an AGP alternative (non-peptide mimics of cathelicidins): ceragenins (CSA-44), on the structure and function of the broiler chicken cecal microbiota. The authors have used next generation sequencing and 1H-Nuclear Magnetic Resonance Spectroscopy (NMR)-based metabolomics and bioinformatics analysis to support their conclusions. This is a very interesting topic and worth of investigation. However, still, I think this manuscript needs a major revision. The experimental design is not acceptable because the number of replicates is too low to derive conclusions from this study: "two replicates in each run (2 runs), 48 animals in total". Results are well explained, and tools utilized for figures and bioinformatics analyses are accurate. My major concern is weak experimental design for the bird trial. Statistical analyses of feed consumption and bird growth average over time is not likely accurate in a trial of 48 birds. What is statistical power of your bird trial? Experimental design i.e., a trial with 48 birds is not feasible to establish a statistical model. Sample size is too small to achieve statistical significance. I suggest increasing number of birds in each treatment.
Author response. We thank the reviewer for their comments. We agree that with respect to ascertaining growth promotion over the duration of the production cycle our study design has limitations, and we have added further information to the discussion section in this regard. Despite the limitations of the study (e.g., sample size and short duration of the study), importantly the manuscript contains novel information that we feel should be available to the scientific community. For example, neither of the antimicrobial growth promoters (AGPs) tested, including virginiamycin, changed the structure of the microbiota, but did alter the function of the microbiota as a function of bacterial community diversity. Moreover, the use of host-defense peptides (HDPs), including HDP mimics is an active area of research in human medicine, and the manuscript examined the impact of a novel HDP mimic as a potential growth promoter. In summary, we feel that our manuscript contains information that will inform research done on this area, and thus should be available to the scientific community.
It was not possible to conduct power analyses as the level of variation to be encountered was unknown. In light of the concern expressed by the reviewer regarding the small number of replicates for the growth promotion metrics, we have re-analyzed the data with eight replicate chicks (i.e., we removed the endpoint variable and analyzing data averaged over days). Following is the degrees of freedom table for the revised growth promotion analyses.
|
Source |
df |
|
AGP (A) |
2 |
|
MT (M) |
1 |
|
A*M |
2 |
|
Error |
42 |
|
Total |
47 |
Specific comments
1. Reviewer comment. Also, I think AGP at low levels are known to increase body weight, however in Line 37 authors mentioned that neither virginiamycin nor CSA-44 influenced weight gain or feed consumption". Please comment.
Author response. In the discussion section 4.1 we have provided the different reasons why we believe we did not observe an improvement in body weight gain due to the administration of AGPs. Moreover, we have added additional information on the limitations of our study that limited our ability to detect subtle differences in growth among the three AGP treatments.
2. Reviewer comment. Any data on use of ceragenins in animal studies? Please comment if its toxicity, and other activities have been evaluated in living organisms? If yes, what levels were effective. Add more information on ceragenins.
Author response. Ceragenins have been tested in different animal species, but not in chickens to our knowledge. One of these studies demonstrated their antimicrobial effect against Clostriudium difficile in mice (Wang et al. 2018, reference [20] in the revised manuscript). Additionally, a study in rats with CSA-44 was conduced by Paul Savage and colleages to specifically evaluate the toxicity of the molecule. No toxicity at the maximum dose evaluated (i.e., 600 mg/kg/day) was observed (Paul Savage unpublished information).
3. Reviewer comment. Line 44-45: Need more evidence to support this conclusion.
Author response. We have tempered the sentence by adding “and this possibility should be examined in future research”.
Reviewer 4 Report
Line 41: How did Glycolysis-related metabolites and amino acid synthesis pathways impacted by virginiamycin and CSA-44?
Line 81: Reward " ------ diseases, leading to increased price of meat.
Line 109: remove "also"
Line 111: change was to were
Line 120: is 4 birds (replicates) per treatment 48 animals total enough sample size? Consider doing power analysis on the number of chicks required.
Line 126: remove chicks "were"
Author Response
Reviewer #4
Author general comment. We appreciate the time expended by the reviewer and the constructive comments provided to improve the manuscript.
Specific comments
1. Reviewer comment. Line 41: How did Glycolysis-related metabolites and amino acid synthesis pathways impacted by virginiamycin and CSA-44?
Author response. We have added “As revealed by metabolomics…” to clarify this statement.
2. Reviewer comment. Line 81: Reward " ------ diseases, leading to increased price of meat.
Author response. The sentence has been modified as suggested.
3. Reviewer comment. Line 109: remove "also"
Author response. The sentence has been modified as suggested.
4. Reviewer comment. Line 111: change was to were
Author response. The sentence has been removed.
5. Reviewer comment. Line 120: is 4 birds (replicates) per treatment 48 animals total enough sample size? Consider doing power analysis on the number of chicks required.
Author response. It was not possible to conduct power analyses as the level of variation to be encountered was unknown. In light of the concern expressed regarding the small number of replicates for the growth promotion metrics, we have re-analyzed the data with eight replicate chicks (i.e., we removed the endpoint variable and analyzing data averaged over days). Following is the degrees of freedom table for the revised growth promotion analyses.
|
Source |
df |
|
AGP (A) |
2 |
|
MT (M) |
1 |
|
A*M |
2 |
|
Error |
42 |
|
Total |
47 |
Despite the limitations of the study (e.g., sample size), importantly the manuscript contains novel information that we feel should be available to the scientific community. For example, neither of the antimicrobial growth promoters (AGPs) tested, including virginiamycin changed the structure of the microbiota, but did alter the function of the microbiota as a function of bacterial community diversity. Moreover, the use of host-defense peptides (HDPs), including HDP mimics is an active area of research in human medicine, and the manuscript examined the impact of a novel HDP mimic as a potential growth promoter. In summary, we feel that our manuscript contains information that will inform research done on this area, and thus should be available to the scientific community. We have added text to the discussion section on the limitations of our study.
6. Reviewer comment. Line 126: remove chicks "were"
Author response. The sentence has been modified to include additional information on transport of chicks and their placement in individually ventilated cages.
Round 2
Reviewer 3 Report
Revised experimental design and statistics are satisfactory.
Reviewer 4 Report
Thanks for revising the manuscript.